# Intrinsic Disorder-Based Design of Stable Globular Proteins

**DOI:** 10.3390/biom10010064

**Published:** 2019-12-30

**Authors:** Galina S. Nagibina, Ksenia A. Glukhova, Vladimir N. Uversky, Tatiana N. Melnik, Bogdan S. Melnik

**Affiliations:** 1Institute of Protein Research, Russian Academy of Sciences, 142290 Pushchino, Moscow Region, Russia; galina-nagibina@rambler.ru (G.S.N.); gkseniya@gmail.com (K.A.G.); tmelnik@vega.protres.ru (T.N.M.); 2Department of Molecular Medicine and Byrd Alzheimer’s Research Institute, Morsani College of Medicine, University of South Florida, Tampa, FL 33620, USA; vuversky@health.usf.edu; 3Institute for Biological Instrumentation of the Russian Academy of Sciences, Federal Research Center “Pushchino Scientific Center for Biological Research of the Russian Academy of Sciences”, 142290 Pushchino, Moscow Region, Russia

**Keywords:** protein stability, intrinsically disorder propensity, stable mutant proteins, design of disulfide bond, design of protein circular permutant

## Abstract

Directed stabilization of globular proteins via substitution of a minimal number of amino acid residues is one of the most complicated experimental tasks. This work summarizes our research on the effect of amino acid substitutions on the protein stability utilizing the outputs of the analysis of intrinsic disorder predisposition of target proteins. This allowed us to formulate the basis of one of the possible approaches to the stabilization of globular proteins. The idea is quite simple. To stabilize a protein as a whole, one needs to find its "weakest spot" and stabilize it, but the question is how this weak spot can be found in a query protein. Our approach is based on the utilization of the computational tools for the per-residue evaluation of intrinsic disorder predisposition to search for the "weakest spot" of a query protein (i.e., the region(s) with the highest local predisposition for intrinsic disorder). When such "weakest spot" is found, it can be stabilized through a limited number of point mutations by introducing order-promoting residues at hot spots, thereby increasing structural stability of a protein as a whole. Using this approach, we were able to obtain stable mutant forms of several globular proteins, such as Gαo, GFP, ribosome protein L1, and circular permutant of apical domain of GroEL.

## 1. Introduction

Elevating protein stability could seem quite a simple and trivial task at first glance, since the interactions and forces responsible for protein stability are well-known from the textbooks [1]. Many examples of protein stabilization via introduction of additional disulfide bond(s) are widely described in the literature [2,3,4,5]. Methods for the selection of stable mutant proteins after random mutagenesis are developed [6,7,8]. For some proteins, stabilizing substitution can be found out by comparison of their amino acid sequences with their homologs from thermophilic organisms [9]. 

However, detailed study of literature makes it clear that in many cases, the stabilizing substitutions of amino acid residues represent the results of long-term search and brute-force selection of various substitution variants, whereas in other cases it is a simple luck, a success apparently brought by chance, or a concourse of favorable circumstances. In other words, currently there are no universal "receipts", which can help an experimenter to elevate the stability of any globular protein by designing an amino acid substitution. However, a general plan of actions might exist. It is known that a protein consists of different structural elements, the sequence of unfolding of which is associated with their stability [1]. The weakest (unstable) structural elements are destroyed first. Intuitively, the targeted stabilization of such weak elements will most effectively affect the overall protein stability. Although the idea is attractive, finding such a weak element in a query protein is not a simple task. We hypothesized that the search for weakened parts within protein structures can be performed using the algorithms predicting intrinsically disorder regions [10,11,12]. 

Programs, such as PONDR^®^ VLXT [13], PONDR^®^ VSL2 [14], and IUPRed [15], as well as PONDR^®^ FIT [12] and IsUnstruct [16], which we used in the current work, utilize protein amino acid sequence to predict the tendency of certain regions of a polypeptide chain of the protein to be either structured or intrinsically disordered based on statistics or properties of amino acid residues. The performance of such programs can be compared to the performance of algorithms predicting secondary structure, because they need only amino acid sequence, and the result of their processing of amino acid sequence is assignment of the intrinsic disorder propensity (IDP) in the (0, 1) range per each residue. When analyzing amino acid sequence of any structured globular protein with known three-dimensional structure by programs such as PONDR^®^ FIT (or IsUnstruct, PONDR^®^ VLXT, PONDR^®^ VSL2, and IUPRed), an interesting feature could be observed: Although the programs classify some protein regions as intrinsically disordered, these regions are ordered in protein structure known from X-ray diffraction analysis. Figure 1 shows the structure of several globular proteins, where the regions predicted by PONDR^®^ FIT as intrinsically disordered are colored blue. It can be seen that according to the X-ray diffraction analysis, these sites are structured, with some of them being α-helices and β-strands of proteins, and not just loops, which one would expect to have increased mobility or a tendency to be in a disordered state. We hypothesized that regions predicted as intrinsically disordered by PONDR^®^ FIT are the regions that require interaction with the remaining rigidly packed part of the protein to obtain the correct three-dimensional structure. We called these regions "weakened", and thus they could be most probably stabilized by introduction of additional disulfide bond, which could lead to stabilization of protein as a whole. 

The current study is dedicated to the experimental examination of the aforementioned hypothesis. For this purpose, our data on stabilization of GFP, L1 family proteins, Gαo, and the data on the design of a stable circular permutant of GroEL apical domain are presented. 

## 2. Materials and Methods 

### 2.1. Protein Expression and Isolation

All the plasmids containing GFP-cycle3 gene (pGFP–cycle3), Gαo from *Drosophila melanogaster* gene (pQE30-Gαo), AaeL1 gene from *Aquifex aeolicus* (pET11a-PL–AaeL1), and HmaL1 gene from *Haloarcula marismortui* (pET11a–PL–HmaL1), were expressed in *E. coli* cells. Plasmids with the mutant genes were constructed by a standard PCR technique using appropriate primers and pET vector as a template. The DNA sequences of all constructs were confirmed by the DNA sequence analysis. All wild type proteins and their mutant forms were expressed and isolated as described elsewhere for GFP–cycle3 [17,18], for Gαo [19,20], for AaeL1 and HmaL1 [21]. The purity of isolated proteins was checked by SDS polyacrylamide gel electrophoresis. 

Recombinant apical domain of GroEL chaperonine from *Thermus thermophilus* (GrAD) and its permutant form were isolated as described in the previous work [22,23].

### 2.2. Protein Chemistry

Protein concentration was determined by UV absorption at 280 nm with extinction coefficients A0.1% 280 = 0.77 [17] for GFP-cycle3, A0.1% 280 = 0.8 for Gαo [19], A0.1% 280 = 0.59 for AaeL1, A0.1% 280 = 0.176 for HmaL1 [21], A0.1% 280 = 0.287 for GrAD [23]. 

Disulfide bond formation for mutant forms of GFP-cycle3, of AaeL1 and of HmaL1 was performed as follows. The pure protein was precipitated by 80% ammonium sulfate. The pellet was resuspended in 0.2 M Tris–HCl, pH 8.8, 0.2 M NaCl, 1 mM EDTA to a protein concentration of 3 mg/ml. The protein was oxidized by addition of oxidized and reduced glutathione to final concentrations of 10 and 2 mM, respectively. After 24 h incubation at room temperature, the glutathione was removed with a PD-10 desalting column. Then, quantity of free SH groups were defined by Ellman’s reagent [24]. Formation of a disulfide bond in the mutant form of Gαo occurred spontaneously in the buffer at pH 7.5. There are 10 free SH groups on the surface of the Gαo protein, to avoid the formation of intermolecular cysteine bridges the mutant protein of Gαo was not oxidized by addition of glutathione. Due to a large number of free SH groups in this protein, we could not use the Ellman^’^s reagent to verify the formation of an SS-bridge. In this case formation of a disulfide bond was checked by SDS polyacrylamide gel electrophoresis because Stokes radii in an unfolded protein with an SS-bridge and without it differ [20]. SDS-PAGE was performed according to Laemmli [25], without addition of reducing agents. 

Completely reduced forms of the mutant variants of all proteins were prepared by incubation with 10 × molar excess of DTT for 30 min at 37 °C in the Tris-HCl buffer at pH 8.5, and then dialyzed against work buffer, 1 mM DTT, pH 7.5 (DTT was left in solution to prevent SH groups from autoxidation). Modified form mutant protein was performed as follows. Free cysteines of reduced form of protein were blocked with 100 mM iodoacetamide for 2 min at 25 °C, pH 7.5.

### 2.3. Search for the “Weakened” Regions in the Amino Acid Sequence of Proteins

To search for the “weakened” regions in proteins, we used programs PONDR^®^ FIT and IsUnstruct [10,11,12] (http://www.pondr.com; http://bioinfo.protres.ru/IsUnstruct). These algorithms use different approaches to predict the presence of intrinsically disordered regions in the protein. 

PONDR^®^ FIT is based on the analysis of a massive set of amino acid sequences of proteins with experimentally validated intrinsically disordered regions. PONDR^®^ FIT is a meta-predictor [12,26] that combines outputs of six individual predictors, PONDR^®^ VLXT [13], PONDR^®^ VSL2 [14], PONDR^®^ VL3 [27], FoldIndex [28], IUPred [15], and TopIDP (PMID: 18991772). Although components of PONDR^®^ FIT have their own advantages and disadvantages (e.g., PONDR^®^ VLXT is not very accurate but shows high sensitivity to local sequence features, whereas PONDR^®^ VL3 is more appropriate for prediction of long disordered regions; PONDR^®^ VSL2 is a rather accurate predictor of both short and long disordered regions, but it has a tendency to overpredict disorder content; IUPred is based on the estimation of pairwise energy content from the amino acid composition; FoldIndex predicts intrinsically disordered regions based on the average residue hydrophobicity and net charge of the sequence; and TopIDP is based on the amino acid scale measuring propensity for intrinsic disorder, this meta-predictor is relatively more accurate than each of the component predictors [12,27], and predicts disordered regions of the protein chain based on the physicochemical properties of amino acids in proteins, such as local amino acid composition and hydrophobicity. For an amino acid sequence, PONDR^®^ FIT outputs are per-residue predictions of the intrinsic disorder propensity in (0, 1) interval. These outputs are then compared to the threshold, and residues with a prediction value greater than the threshold (0.5) are predicted to be intrinsically disordered. 

The algorithm of the IsUnstruct program [10] is based on the Ising model for prediction of disordered residues from protein sequence alone. According to this model, each residue can be in one of two states: Ordered or disordered. The model is an approximation of the Ising model in which the interaction term between neighbors has been replaced by a penalty for changing between states (the energy of border). The authors analyzed crystalline structures of proteins from PDB, having chosen such proteins that have polypeptide chain regions in their structure not resolved yet by the X-ray structural analysis.

### 2.4. Circular Dichroism

The circular dichroism (CD) spectra and ellipticity dependence on temperature were measured using a JASCO–600 spectropolarimeter (Japan Spectroscopic Co, Japan) equipped with a temperature-controlled holder in 0.185 mm thick cells at the protein concentration of 0.3–0.5 mg/ml. The molar ellipticity [θ] was calculated from the equation, [θ] = θ_obs_*M_res_/C*L where C is the protein concentration (g/l), L is the optical path length of the cell (mm), θ_obs_ is the ellipticity measured (in degrees) at wavelength λ (nm), and M_res_ is the mean residue molecular mass of the protein. The dependence of ellipticity at 222 nm for Gαo and GrAD, at 215 nm for AaeL1 and HmaL1, on temperature was determined with a heating rate of 1 K/min.

### 2.5. Differential Scanning Calorimetry (DSC)

DSC studies were conducted on a SCAL-1 differential scanning microcalorimeter (Scal, Russia) using a glass cell with a volume 0.33 mL at scanning rates 1.0 K/min [29]. The measurements were conducted at 2 atm pressure. Data collection was carried out automatically with a temperature step of 0.1 K. The Wscal program developed in the Laboratory of Protein Thermodynamics (Institute of Protein Research, Russian Academy of Sciences, Pushchino) was used for data recording. The protein concentration in microcalorimetric experiments was 1 mg/mL. In order to obtain values of the protein partial heat capacity, the SCAL program was used, which is based on the algorithms described by Privalov and Potekhin [30].

## 3. Results and Discussion

### 3.1. Analysis of Literature on Protein Stabilization

At first glance, it seems that in order to confirm or refute the aforementioned hypothesis, it would be sufficient to analyze the available literature data. For example, one might choose publications, where the authors stabilized proteins by introducing artificially engineered disulfide bonds. Then, via using programs such as PONDR^®^ FIT, one can find parts of the polypeptide chain that are weakened (predicted as intrinsically disorder) and try to answer the question if there is really a correlation between the stabilization of the protein and the location of the mutation in the sites that are predicted as intrinsically disorder? However, it turned out that the literature data analysis would not help in finding such a correlation, since many factors influence the stability of proteins with artificially introduced disulfide bonds, such as the size of the protein, the specific features of its structure, as well as the design features of disulfide bonds. For example, the introduced disulfide bonds almost always stabilize small proteins, regardless of their location (see [4,31]). In addition, when comparing the effects of different disulfide bonds on protein stability, it is important for the artificial disulfide bonds to be equally designed. For example, it is not correct to compare the stabilizing effects of a disulfide bond introduced on the surface of a protein and that is placed in its hydrophobic core. It is clear that the disulfide bonds within the hydrophobic core of the protein could lead to either destabilization of a protein (if it breaks the packing of the protein hydrophobic core) or to protein stabilization, if it stably joints different structural parts of a protein. Similar reasoning leads to the idea that it is not quite correct to compare the effect of disulfide bonds introduced in different parts of a large protein, if secondary structure of these parts is very different.

Using the example of several proteins below, we have tried to demonstrate why it is impossible to use literature data to find a correlation between protein stabilization and the location of the disulfide bonds (Figure 2). At first glance, these studies confirm our hypothesis. However, if we consider the arguments about the influence of different factors on protein stabilization given above, it turns out that no reliable confirmation for the hypothesis can be drawn. Figure 2 shows the PONDR^®^ FIT-based plots of the predicted intrinsic disorder propensities (PIDP) within the amino acid sequences of several such proteins. In addition to these disorder plots, Figure 2 shows the positions of cysteine residues introduced by the authors of the cited works and the corresponding elevation or diminution of stability temperature of the protein in degrees as digits. From Figure 2A it could be seen that the N-terminus of the lysozyme can be regarded as the weakest spot (PIDP > 0.5) of this protein. This is possibly the reason of why the three introduced disulfide bonds "fixing" or “stitching” the N-terminus to the protein globule do stabilize the protein [2]. On the contrary, two disulfide bonds inserted into the well-structured protein region (residues 100–150 with PIDP < 0.5) do not stabilize the protein (Figure 2A). It would seem that the work on the stabilization of lysozyme [2], confirms our assumption. However, perhaps such a different effect of the introduced disulfide bonds on the stability of lysozyme is due to their location in the two domains of lysozyme. Three disulfide bonds that stabilize the protein connect the N- and C-terminal domains of lysozyme. Two disulfide bonds that destabilize the protein are located within the same C-terminal domain.

We must also comment on the results of introduction of disulfide bonds into small proteins (Examples on Figure 2C,D [32,33]). Disulfide bonds in any part of a small (around 100 amino acid residues) protein are expected to lead to the overall stabilization of protein structure. The reason is that most often, small proteins fold/unfold without intermediate states. On the other hand, introduction of disulfide bonds in any part of a small protein leads to the sizable compactization of the unfolded state, thereby stabilizing the native state of the protein. It should be reminded here that the free energy of protein stabilization consists of entropy and enthalpy components. For small two-state proteins, it is shown that the gain of free energy when introducing disulfide bonds is associated mainly with a decrease in the entropy of the unfolded state of the protein [31]. That is why, for example, both disulfide bonds in immunoglobulin domain [33] resulted in the protein stabilization (Figure 2D). This is in contrast to the perspectives of our hypothesis, since the disulfide bonds introduced into the middle part of the protein (residues 50–70) should not lead to sizable stabilization, since it falls into the region with low intrinsic disorder propensities (Figure 2D).

Folding of large proteins is accompanied by the formation of several intermediate states, and large proteins contain several structural elements (domains) with different stability. Therefore, for such proteins, it is very important to know in which structural element the disulfide bond is introduced. By stabilizing the weakest of these elements, one can stabilize the protein as a whole. Figure 2B shows a PONDR^®^ FIT-based plot of the predicted per-residue intrinsic disorder propensity of neutral protease. It can be seen that this protein has two regions with high PIDP values, the N-terminus of the protein and the region around residues 200–250. A disulfide bond introduced at the N-terminus of the neutral protease led to protein stabilization (Figure 2B) [34]. However, confirmation of our hypothesis would require testing the effects of the disulfide bonds introduced into other regions of the protein. Unfortunately, there is no such data.

An interesting result was reported in the work on the engineered disulfide-based stabilization of subtilisin [35]. No one of the studied in the work substitutions led to stabilization of the protein [35]. On one hand, all the residues chosen for substitutions were located within the protein regions with rather low PIDP (<0.5, Figure 2E), confirming our hypothesis. However, on the other hand, other factors could also contribute to these outputs, such as the presence of some specific structural features of the protein. Subtilisin is a unidomain protein with complex structure, where the elements of secondary structure are packed into four layers [36] (PDB:2ST1). Therefore, the majority of the disulfide bonds introduced by the authors “fix” or ”stitch” together the elements of secondary structure located within the core of the protein, that are clamped by other structural elements of subtilisin. 

The aforementioned examples show that the literature data do not allow us to unequivocally confirm or refute our hypothesis, since many factors might affect stability of proteins with the introduced disulfide bonds, such as design features of cysteine bridges, structural features of target proteins, and their sizes. Therefore, a validation of our hypothesis can be conducted either using bioinformatics methods—by collecting a large database of proteins with introduced disulfide bonds and finding correlations between stability and IDP, or experimentally. We decided to go for an experimental test of our hypothesis, systematically excluding factors associated with the structural features of proteins and the design features of disulfide bonds. There are at least two ways to do this: (1) Select one protein that has similar repeating structural elements, calculate PIDP and introduce a disulfide bond into each of the structural elements, and then evaluate the effect of mutations on protein stability; (2) select two proteins with different amino acid sequence, but with the same three-dimensional structure and carry out similar studies.

### 3.2. Investigation of Green Fluorescent Protein

We have been studying the folding features of green fluorescent protein for a long time [18,37,38,39,40,41,42]. This protein is a β-barrel consisting of very similar β-strand (Figure 3). Therefore, in its structure, one can find similar structural elements and select pairs of close amino acid residues and design disulfide bonds on the surface of the protein (to avoid disruption of the internal packaging of the protein).

Figure 3A,B show two projections of the GFP β-barrel, whereas Figure 3C–F represent structural elements selected for the experimental analysis, and Figure 3G shows the plot of intrinsic disorder propensities for amino acid sequences of green fluorescent protein calculated by PONDR^®^ FIT and IsUnstruct. 

Since these two programs use absolutely different approaches for calculation of the intrinsic disorder propensities (See Materials and Methods), from our point of view, comparison of the results of PONDR^®^ FIT and IsUnstruct should help in minimization of the program selection-dependent errors and select the weakened regions of the proteins more reliably. Figure 3G shows that the only large protein region predicted as intrinsically disordered (PIDP > 0.5) by both programs is located at the C-terminal part of GFP. Therefore, this C-terminal region could be regarded as weakened. GFP comprises a "barrel" composed of similar β-hairpins. To check our hypothesis, we decided to cross-link both the predicted weakened region (which is a hairpin) and other similar beta-hairpins of the protein with a disorder score below 0.5 by disulfide bonds. Figure 3 shows 3D structure of GFP, separate view of β-hairpins, and amino acid residues substituted to cysteines. In this study, four mutant variants with substitutions V11C–D36C, Q111C–V93C, K162C–Q184C, S202–T225C were investigated [18].

Figure 4 shows the equilibrium urea-induced unfolding curves of the wild type GFP and its four double-cysteine mutants in oxidized forms (V11C–D36C, Q111C–V93C, K162C–Q184C, S202C–T225C) monitored by changes in far-UV CD spectra. It can be seen that only one of the introduced disulfide bonds stabilized the protein. This was the S202C–T225C mutant form. In the previous works [18,39,40,41], we conducted DSC experiments and described heat denaturation of wild type GFP and its mutant oxidized and modified forms. In these previous studies, thermal denaturation of GFP was shown to be quite a complex non-equilibrium process that could be described by a model including two consequent irreversible denaturation stages. Nonetheless, those DSC studies allowed us to conclude that only one mutation, S202C–T225C, stabilized the protein, whereas other mutations destabilized the protein, affecting activation energies and rate constants of heat denaturation. 

Therefore, based on these observations we can conclude that the introduction of disulfide bonds into weakened region of GFP that are predicted by PONDR^®^ FIT and IsUnstruct programs as intrinsically disordered leads to stabilization of the protein.

Another confirmation of our hypothesis that the use of the predictors of intrinsic disorder to find regions with high PIDP in structured proteins can help in discovering their weakened spots was found in ribosome proteins L1 from different organisms.

### 3.3. L1 Protein Study

To find convincing confirmation of correlation of high intrinsic disorder propensities with weakness of protein tertiary structure one must attempt to exclude all other factors affecting protein stability during introduction of cysteine bridges, such as structural features of the proteins and positions of amino acid residues specific for mutations.

The idea is that to correctly test our assumption, it is necessary to study two proteins with identical three-dimensional structures, but quite different amino acid sequences. Similar three-dimensional structure gives us the opportunity to introduce amino acid substitutions into similar regions of proteins with identical secondary structure. Different amino acid sequences give different intrinsic disorder propensity values for structurally identical parts of the proteins. Hence, introduction of identical substitutions into similar structural motifs of proteins with various intrinsic disorder propensities allows us to exclude the effect of secondary structure context for the regions of introduced disulfide bonds.

Two ribosomal proteins L1 were chosen for this study, L1 from the halophilic archaeon *Haloarcula marismortui* (HmaL1) and L1 from the extremophilic bacterium *Aquifex aeolicus* (AaeL1). The three-dimensional structure of AaeL1 is known from the experiments [43]; for HmaL1, a homology model was built [44]. Both proteins have highly conservative three-dimensional structure specific for ribosome L1 proteins, which can be divided into two domains. The amino acid sequences of these proteins have sequence identity of only 33%. Hence, this pair of proteins satisfies our requirements: They are similar in structure, but quite different in sequence.

#### 3.3.1. Comparison of Spatial Structure and Amino Acid Sequences of AaeL1 and HmaL1 Proteins

The three-dimensional structure of L1 proteins is remarkably conservative. It can be divided into two domains (I and II) connected by a flexible linker allowing to change mutual orientation of the domains. There are two variants of domain orientation in the solution, "open" and "closed" states [21].

Figure 5 (left) shows superposition of the experimentally determined 3D structure of AaeL1 (blue) and modelled 3D structure of HmaL1 (green). Both proteins have two-domain structure specific for L1 proteins: N- and С-tails are located within domain I, which is connected by two-strand linker with domain II. The basic elements of secondary structure of domain I in these two proteins are superposed quite well onto each other. When comparing two protein structures, a specific difference between archaeal and bacterial L1 proteins can be seen; i.e., different rotation degree of the domain II relative to domain I. However, when comparing only domain II structures, we do also see good superposition (see Figure 6 right).

Alignment of amino acid sequences of the proteins in BLAST (https://blast.ncbi.nlm.nih.gov/Blast.cgi) showed 33% amino acid identity between AaeL1 and HmaL1. Figure 5 (right) shows alignment of amino acid sequences of two proteins, the identical amino acid residues are colored gray. AaeL1 and HmaL1 differ in their length: AaeL1 consists of 242 a.a., HmaL1 consists of 212 a.a.

#### 3.3.2. Mapping of Disordered Regions and Design of Disulfide Bridges in AaeL1 and HmaL1

Disordered regions in AaeL1 and HmaL1 proteins were mapped using PONDR^®^ FIT and IsUnstruct programs [10,12]. Figure 6 shows intrinsic disorder propensities for amino acid sequences of AaeL1 (blue) and HmaL1 (green). Gray square shows the region of protein sequence belonging to domain II. In this domain, we see large difference between AaeL1 and HmaL1 proteins in their intrinsic disorder propensities (the region is underlined by red line below the plot). For AaeL1 protein (Figure 6 left, blue line), this region is predicted as structured (low PIDP values), whereas for HmaL1 (Figure 6 left, green line) it is predicted as intrinsically disordered (high PIDP propensity). Therefore, this is a region most suitable for the examination of our idea. If the results of PONDR^®^ FIT and IsUnstruct can be really interpreted as prediction of structured and weakened regions of polypeptide chain, the introduction of disulfide bonds into a region predicted as disordered in HmaL1 should lead to elevated stability and decreased conformational mobility of this protein, whereas insertion of disulfide bonds into the same region predicted as structured in AaeL1 would not affect protein stability or would even decrease it. If local spatial structure is more significant for protein stabilization, then the introduced replacements should affect both proteins similarly.

It is worth noting that a detailed analysis of Figure 6 left allows us to understand why it is difficult to select a pair of proteins with the same three-dimensional structure but different IDPs to test our hypothesis. We examined several pairs of proteins, including ribosomal proteins from different organisms. However, it turned out that either the PIDP values calculated for different proteins are close to each other, despite the different amino acid compositions (this is evident, for example, for the regions around the residues 150 or 200, Figure 6 left), or the predictions of different programs are very different. For example, in Figure 6 left the PIDP, calculated PONDR^®^ FIT and IsUnstruct differ in the region around the residue 25 for AaeL1, and in the region around residue 50 for HmaL1.

Figure 6 (right) shows superposition of the three-dimensional structure of domains II of AaeL1 and HmaL1 proteins. The average RMSD calculated from C_α_ atoms is 2.8 Å. Blue and green colors show the positions of regions chosen for introduction of disulfide bonds based on the predictions of PONDR^®^ FIT and IsUnstruct, respectively. Yellow spheres show the residues chosen for substitution to cysteine residues, D101 and K127 in AaeL1, and E82 and D114 in HmaL1 protein. Selection of amino acid residues for substitution to cysteines was made based on two criteria: 1) Distance between C_β_-atoms of amino acid residues must be ~5 Å; 2) the residues must point to each other in space.

#### 3.3.3. Investigation of the HmaL1–E82C-D114C Conformational Stability 

Heat denaturation of the wild type HmaL1 and HmaL1–E82C–D114C mutant form was studied by CD spectroscopy. Figure 7 shows the dependence of the fraction of native state of wild type HmaL1 and a mutant with inserted disulfide bonds, HmaL1–E82C–D114C, and modified by iodoacetamide, HmaL1–E82C–D114C mod, analyzed by changes in molar ellipticity at 215 nm. HmaL1 and HmaL1-E82C-D114C mod are melting at 55 °C, whereas HmaL1–E82C–D114C with an oxidized disulfide bond melts at 65 °C. Therefore, the substitutions for cysteine alone do not contribute significantly to protein stability, whereas the formation of a disulfide bond stabilizes the HmaL1protein. This confirms our suggestion that disulfide bonds inserted into a region predicted to be disordered (despite being structured in an actual protein) should lead to the increase in protein stability.

#### 3.3.4. Investigation of the AaeL1–D101C–K127C Conformational Stability

Heat denaturation of the wild type AaeL1 and its variant with disulfide bond, AaeL1–D101C–K127C, was studied by circular dichroism spectroscopy. One should remember that the AaeL1 protein is from an extremophilic bacterium, and, as a result, it is characterized by a very high conformational stability. Therefore, thermal denaturation of this protein was investigated in the presence of 4 M urea. Figure 8A shows curves of the temperature dependence of the molar ellipticity at 215 nm. It is clear that the curves coincide for the wild type protein and its mutant form with disulfide bond, AaeL1–D101C–K127C. To further validate the melting points of wild type AaeL1 and AaeL1–D101C–K127C, we studied the melting of these proteins by differential scanning microcalorimetry. Figure 8B represents the melting curves of AaeL1 and its oxidized double cysteine mutant form with disulfide bond in the presence of 4 M urea. Melting points coincide for wild type and mutant variants, comprising 347 K. Therefore, it follows from the presented data that the introduction of a disulfide bond into the protein region, which is predicted by the PONDR^®^ FIT and IsUnstruct to be structured, does not lead to protein stabilization.

Therefore, studies on L1 and GFP confirmed our hypothesis that programs calculating intrinsic disorder propensities can be used for mapping the weakened regions of proteins. Hence, they can be used for designing stable proteins.

### 3.4. Stabilization of Gαo

One of the research groups of our institute had a task to obtain crystals of a Gαo protein from *D. melanogaster*, but their multiple attempts were unsuccessful. We suggested to create a mutant of this protein, assuming that the stable form of this protein would be better suitable for crystallization than the wild type protein. Gαo consists of two domains. Similar to ribosomal L1 protein, the first domain is formed by the N– and С–terminal regions of Gαo (1–64 and 181–345), and domain II comprises its central region (65–180) [45]. Domain I of Gαo protein possesses GTPase activity, and its amino acid sequence is conserved among various organisms. The sequence of the second domain varies in different organisms, and this variance can be the reason for instability or mobility of domain II of Gαo from *D. melanogaster*, resulting in inability to crystallize this protein. Therefore, we decided to stabilize the second domain of Gαo protein using the proposed approach for rational design of artificial disulfide bond.

Figure 9 shows intrinsic disorder propensities calculated for amino acid sequence of Gαo. The plots built by PONDR^®^ FIT and IsUnstruct are apparently similar. Vertical dashed lines on Figure 9 border the region of Gαo forming the domain II of this protein (residues 65–180). Both programs used for calculations predict quite high propensity of 90–120 aa region to be in disordered state. We decided to choose exactly this "weakened" region for stabilization by introduction of a disulfide bond.

Design of disulfide bond in Gαo from *D. Melanogaster* and characterization of the resulting constructs are described in detail in our previous study [20]. Figure 10 shows the differential scanning microcalorimetry generated melting curves of Gαo and its mutant form with introduced disulfide bond. Each of melting curves shows two maxima, which, as we showed [20], are related to melting of two domains of Gαo. Figure 10 shows that the position of the first melting peak coincides in wild type protein and its mutant form (T_m1_~320К), whereas the second peak of the melting curve differs in wild type ( T_m2_ = 329 K for wild type Gαo, а T_m2_ = 333 K for the mutant). Therefore, the introduced disulfide bond stabilized one of the Gαo domains and increased its melting point by 4 degrees.

### 3.5. Searching for Stable Circular Permutant Variants of the GroEL Apical Domain

When designing complex fusion proteins, sometimes a task of creation of circular permutants arises. For this purpose, N– and C–termini of the protein are tailored with a linker, and a site for cleavage is selected in a protein structure, defining novel N– and C–termini of the protein. It often leads to dramatic destabilization of the protein. How can one select cleavage site in a protein without affecting its stability? We believe that the aforementioned approach based on the prediction of intrinsic disorder predisposition can be used for this purpose too. Particularly, intrinsic disorder propensities for circular permutants with different cleavage positions should be calculated, and the position resulting in maximal degree of order of the structure should be selected. This sub-project was inspired and initiated by our colleagues who studied apical domain of GroEL chaperone [22,23]. The protein chosen for a practical task to design a stable circular permutant was an apical domain of GroEL. For this purpose, we started with calculation of intrinsic disorder propensity of GroEL apical domain sequences with different positions of N– and C–termini. 

Figure 11A shows intrinsic disorder propensity plots only for three proteins that were isolated and experimentally studied. Gro192 is a wild type GroEL apical domain, Gro207 is a domain with N– and C–termini tailored by a linker and cleavage at the amino acid residue 207, and Gro230 is a circular permutant with cleavage site at position 230. Intrinsic disorder propensities were calculated in analogous manner for all the variants of N– and C–end position with the step of 5 residues. For each of such plots, number of amino acid residues with disorder score above 0.5 can be calculated. For instance, in Gro192, this value equals to 10 for the N–terminal region and 10 for the C–tail; i.e., 20 amino acid residues have an PIDP > 0.5. Analogous calculation for sequences with different cleavage site position allows building the dependency of the number of amino acid residues with intrinsic disorder on the position of cleavage site in circular permutant. An example of such a plot is shown on Figure 11B. A minimum on the plot means the lowest number of residues with IDP > 0.5; i.e., this variant can be regarded as most stable. Maximal value means the highest number of amino acids with IDP > 0.5; i.e., this circular permutant variant would be unstable. According to our calculations, the most stable variant of circular permutant would arise if the cleavage site would be designed at the 207 residue. For additional examination, we designed a circular permutant with cleavage site at the 230 residue. According to the plot on Figure 11B, this variant should be similar to the wild type protein or even slightly destabilized.

Having isolated and purified the apical domain of GroEL and its two circular permutants, we studied their stability. Figure 12A shows circular dichroism spectra and melting curves of the proteins. Far UV CD spectra of Gro192 and Gro207 appear to be almost identical. Their shape and intensity correspond to structured proteins. Shape of the Gro230 far-UV CD spectrum suggests the lowered degree of structure in this protein. Figure 12B shows melting curves for Gro192 and Gro207 proteins. Since Gro230 aggregated during heating, it was impossible to record its melting curve. It can be defined from the plots on Figure 12B that the circular permutant of Gro207 protein is destabilized by only two degrees compared to wild type protein, which can be considered as an undoubted success of the design of this mutant.

The results of the design of circular permutants described above does also confirm our assumption on feasibility of intrinsic disorder propensity predicting programs to search for weakened or stable protein regions, that could be the targets for introduction of diverse mutations.

## 4. Conclusions

In this work, we have successfully used algorithms for the evaluation of intrinsic disorder predisposition (such as PONDR^®^ FIT and IsUnstruct) as tools for searching for the weakened regions in structured globular proteins. Using GFP, GαO, L1 protein, and GroEL apical domain as examples, we have shown that the weakened regions found by these programs as regions with highest levels of predicted intrinsic disorder predisposition are a suitable target for introduction of stabilizing mutations, for example, disulfide bonds. Using L1 ribosomal proteins as the example of proteins with similar tertiary structure but different sequences, we have shown that there is a real correlation between the stabilizing effects of disulfide bonds introduced into weakened protein spots and the intrinsic disorder propensity score of a polypeptide chain, rather than between disulfide bond stabilization and the specific features of polypeptide chain packing. Analysis of the GroEL apical domain has allowed us to show that PONDR^®^ FIT and IsUnstruct can be used as tools to design stable circular permutants of proteins.

## Figures and Tables

**Figure 1 biomolecules-10-00064-f001:**
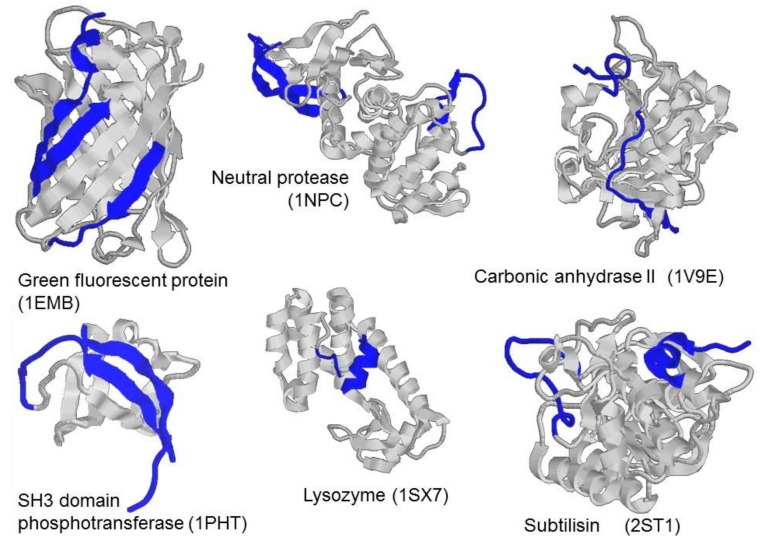
Three-dimensional (3D) structures of green fluorescent protein (1EMB), neutral protease (1NPC), carbonic anhydrase (1V9E), SH3 domain phosphotransferase (1PHT), lysozyme (1SX7), subtilisin (2ST1). Blue color highlights regions predicted by PONDR^®^ FIT program as intrinsically disordered.

**Figure 2 biomolecules-10-00064-f002:**
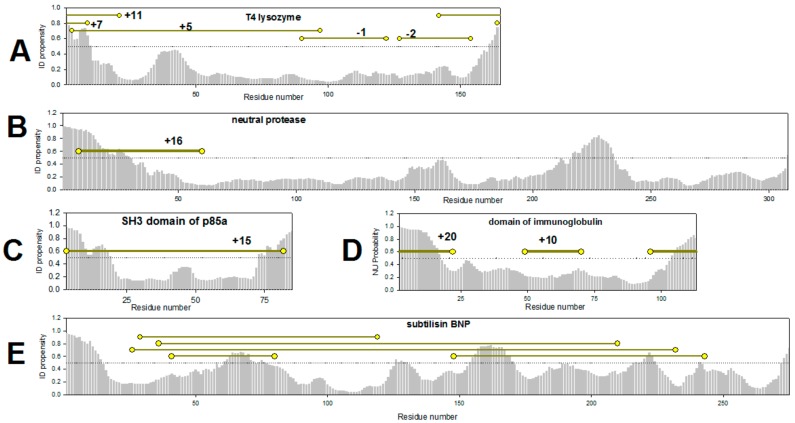
Intrinsic disorder propensity evaluated by PONDR^®^ FIT for the amino acid sequences of T4 lysozyme PDB:1SX7 (**A**), neutral protease PDB:1NPC (**B**), SH3 domain of p85a PDB:1PHT (**C**), immunoglobulin domain PDB:1HCV (**D**), subtilisin PDB:2ST1 (**E**). Yellow circles show the positions of amino acids substituted to cysteine residues for introduction of disulfide bonds [2,32,33,34,35].

**Figure 3 biomolecules-10-00064-f003:**
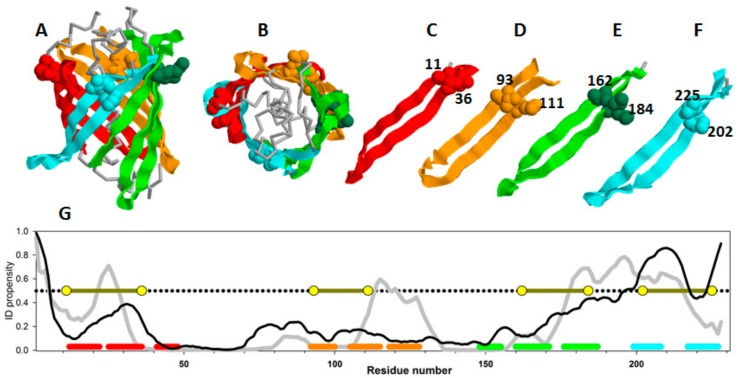
The three-dimensional structure of GFP: Side view (**A**) and top view (**B**), β-strands of the protein colored red, yellow, green, and cyan, respectively. β -hairpins selected for insertion of disulfide bonds are shown separately (**C**–**F**). Amino acid residues substituted to cysteine are shown as spheres their numbers are labeled on the figure. (**G**) ID propensity plots calculated by PONDR^®^ FIT (gray line) and IsUnstruct (black line). The corresponding colored lines at the bottom of the plot highlight the positions of beta-hairpins selected for the study. Yellow circles show the positions of disulfide bond insertions.

**Figure 4 biomolecules-10-00064-f004:**
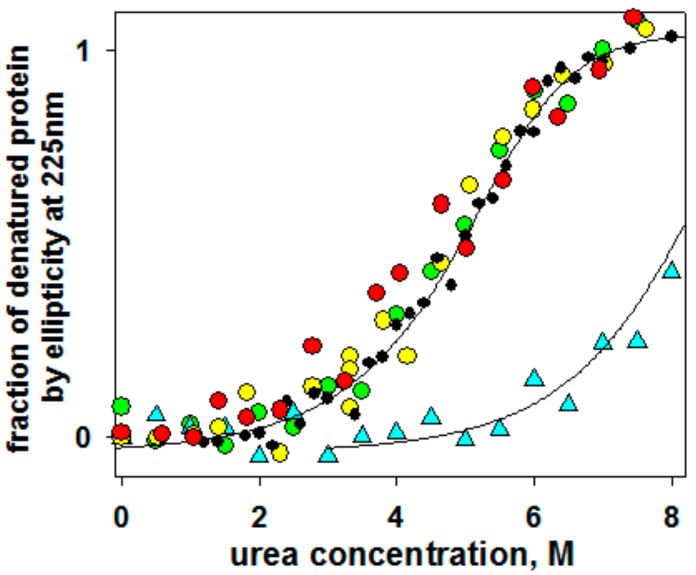
Dependency of fraction of denatured state calculated by ellipticity at 225 nm on urea concentration in protein samples at pH 7.2: Wild type (black circles) and four oxidized double-cysteine mutant forms, V11C–D36C (red circles), Q111C–V93C (orange circles), K162C–Q184C (green circles), S202C–T225C (cyan triangles).

**Figure 5 biomolecules-10-00064-f005:**
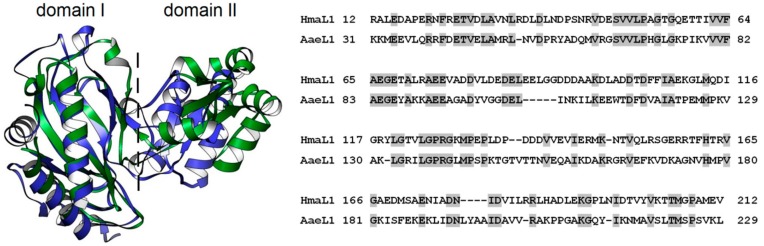
(**Left**) Superposition of three-dimensional structure of AaeL1 (PDB:3QOY) (blue) and modelled of three-dimensional structure of HmaL1 [44] (green). The dashed line separates two domains, I and II. (**Right**) Alignment of ribosome proteins HmaL1 and AaeL1. The identical residues are highlighted in gray.

**Figure 6 biomolecules-10-00064-f006:**
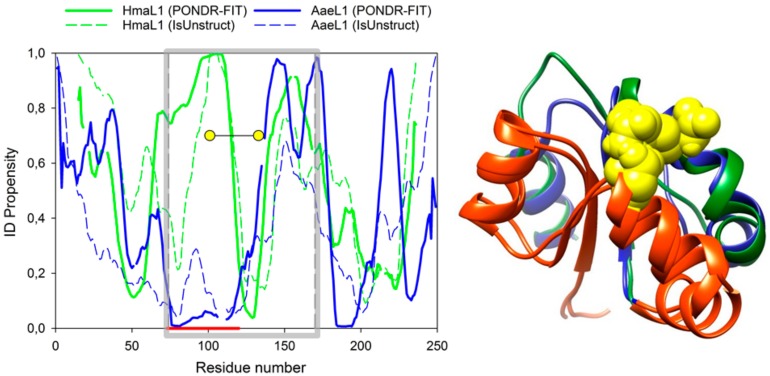
(**Left**) Calculation of intrinsic disorder propensities for amino acid sequences in HmaL1 (green) and AaeL1 (blue) proteins performed in PONDR^®^ FIT (solid lines) and IsUnstruct (dashed lines). Gray square limits regions of amino acid sequences forming domain II. Thick red line below the plot shows a region of the highest difference in intrinsic disorder propensities between AaeL1 and HmaL1 proteins. Yellow circles show the positions of inserted disulfide bonds. (**Right**) Superposition of three-dimensional structure of domains II of HmaL1 (green) and AaeL1 (blue). Regions of the highest difference in IDP part of AaeL1 and HmaL1 is colored red. Yellow spheres highlight amino acid residues selected for substitution to cysteine residues, D101 and K127 in AaeL1 and E82 and D114 in HmaL1.

**Figure 7 biomolecules-10-00064-f007:**
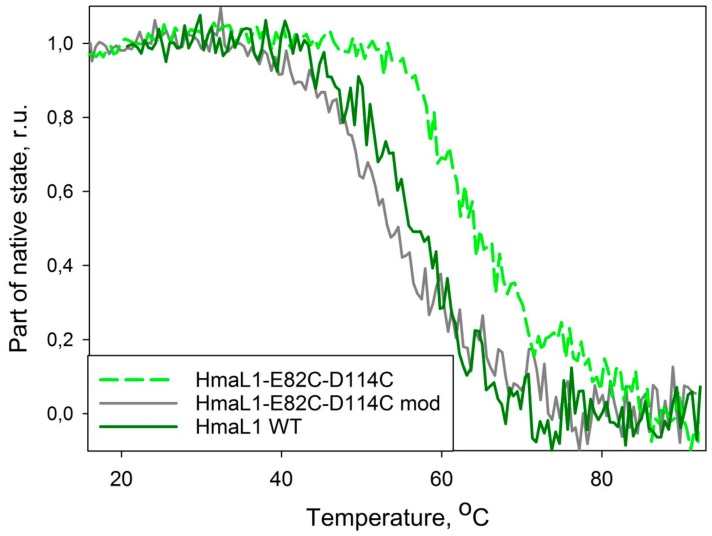
Dependency of fraction of native state of wild type HmaL1 (HmaL1 WT, solid green line), mutant with inserted disulfide bridge (HmaL1–E82C–D114C, dashed light-green line) and mutant form with iodoacetamide-modified sulfide group (HmaL1–E82C–D114C mod, solid gray line) on temperature. Fraction of native protein state was calculated from molar ellipticity at 215 nm wavelength.

**Figure 8 biomolecules-10-00064-f008:**
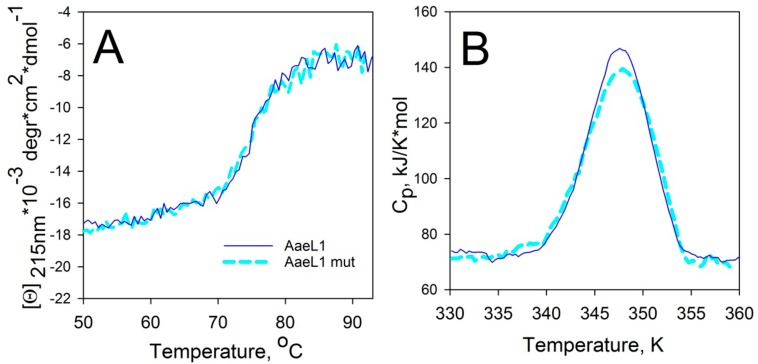
Heat denaturation of AaeL1 proteins (wild type and AaeL1–D101C–K127C). (**A**) Dependence of molar ellipticity at 215 nm on temperature in 4 M urea for wild type AaeL1 (solid line) and for AaeL1–D101C–K127C (dashed line). (**B**) Dependence of partial heat capacity on temperature in 4 M urea for wild type AaeL1 (solid line) and AaeL1–D101C–K127C with oxidized disulfide bond (dashed line).

**Figure 9 biomolecules-10-00064-f009:**
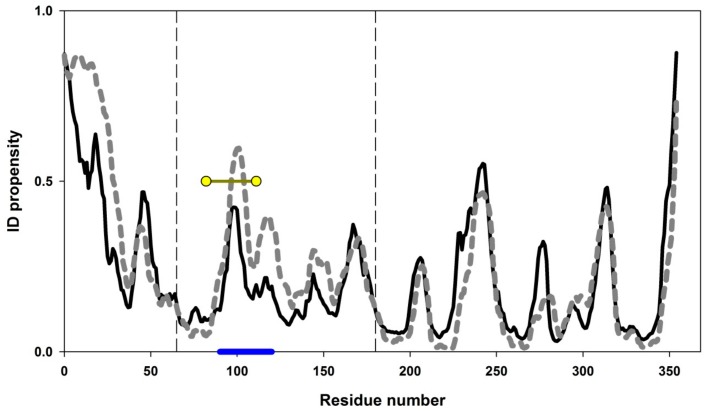
Intrinsic disorder propensities for amino acid sequence of Gαo from *D. melanogaster*. The plots were calculated in PONDR^®^ FIT (solid line) and IsUnstruct (dashed line). Vertical dashed lines highlight the amino acid sequence region (65–180) forming domain II of Gαo. Blue line under the plot shows the region to be stabilized by disulfide bond introduction. Yellow circles show the exact positions of amino acid residues, between which the disulfide bond was designed.

**Figure 10 biomolecules-10-00064-f010:**
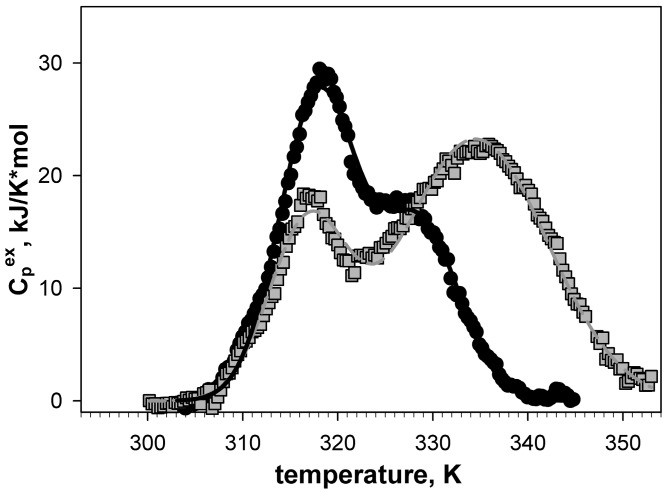
Dependency of excessive heat capacity of wild type Gαo protein (circles) and its V82C mutant with disulfide bond introduction (squares) on temperature [20].

**Figure 11 biomolecules-10-00064-f011:**
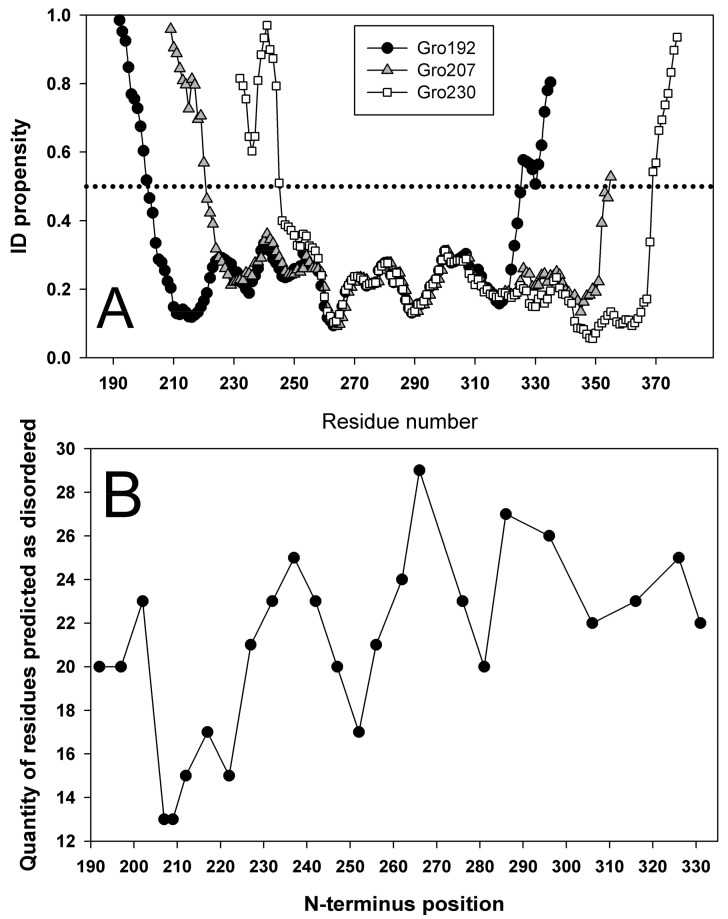
(**A**) Intrinsic disorder propensities (IDP) for amino acid sequence calculated for Gro192, apical domain of wild type GroEL cleaved from GroEL protein by 192 and 335 amino acid residue; Gro207 is a domain with N– and C–termini (a.a. 192, 335) tailored by a linker, and a cleavage (novel N– and C–ends) made at residue 207; Gro230 is a circular permutant with cleavage at a.a. 230. (**B**) Dependency of number of amino acid residues with IDP > 0.5 on the position of cleavage site.

**Figure 12 biomolecules-10-00064-f012:**
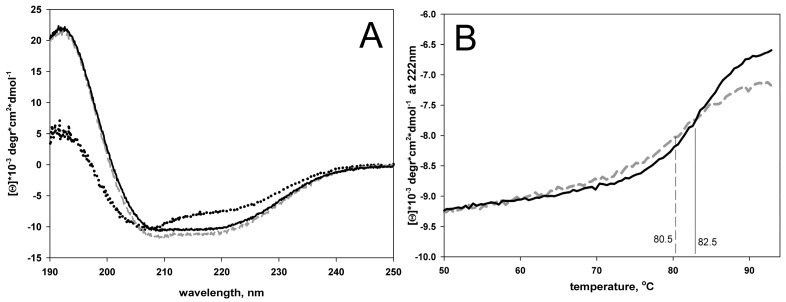
Circular dichroism spectra (**A**) and melting curves (**B**) of Gro190 (solid line), Gro207 (dashed line), and Gro230 (dots on figure A). The centers of the melting curves (melting temperatures) are shown by vertical lines on figure B.

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
