# Peer review of "Intrinsic Disorder-Based Design of Stable Globular Proteins"

_biomolecules, 2019, doi:10.3390/biom10010064_

Round 1

Reviewer 1 Report

This study aims to characterize determinants of protein stabilization by prediction of intrinsically disorder regions in some globular proteins. The approach uses a Circular Dichroism and DSC as main techniques. In addition to both assays, the authors perform denaturation assay in presence of Guanidium Chloride and Temperature influence. Several published work, spanning the last 20 years, have presented evidence consistent with the current manuscript, but nonetheless, this paper adds a little bit more details and tend to clarify some statements that authors assume.
Taken together the results of the paper are needed to be amended for being suitable for publication in Biomolecules. The are some points that should be amended or clarified before the manuscript may be accepted. These are detailed in the following:
Major revisions:

Authors used PONDR-FIT and IsUnstruct for predictions of IDR. First of all, reviewer was not able to find IsUnstruct algorithm as link provided at original paper seems to be broken and after a quick search was not able to find it. Would be great if it could be provided. Authors should explain why they chose the selected globular proteins as it looks an author selection and criteria should be clarified. Finally, would be interesting if hypothesis about secondary/structure could be confirmed by another IDR algorithms as IUpred or similar. Methods and Materials are poor explained and not all the references explain correctly the protocols. Specifically, authors should indicate how they demonstrate that S-S bonds are formed. They indicated by SDS-PAGE. In this case, SDS gels should be provided, in other case, most of the proteins selected presents their own Cysteines in the wild type sequence. An alternative method should check the introduced C-C bond are formed, and cannot discard that new C-C bonds are formed between the Cys Wt and new Cys. In Figure 2B, a discussion about region 200-250 residue would be appreciated as algorithm methods predicted IDR region. The sentences between lane 168-176 and lanes 231-243 are extremely confusing and it is not clear why and why not the algorithm predictions can be supported by the literature or not. Additionally, please provide references that confirms the final hypothesis. Authors indicated that AaeL1 has a PDB structure but HmaL1 is built from a homologous model. It has strong implications for the author hypothesis, as they cannot demonstrate 100% their hypothesis. A sentence clarifying this situation should be stated. For GroEL experiment, they assure that a stabilizing mutation has been introduced but experiments do not confirm that, please provide more explanation about so much affirmative sentences.

Minor revisions:
-

Figure 2 and 6 are little bit confusing. It would be really informative more detailed footnotes or some figure improvement. Lane 33-34 It is not clear the meaning of the sentence. Please, re-phrase or use appropriate concepts. Lane 147-150: Is not clear the assumption without references that “small proteins fold/unfold without intermediate state” Experiments in Figure 8 are performed in 4M Urea, authors should indicate why this denaturant agent is present.

Author Response

Dear editor and reviewers, I and the co-authors of the article are grateful for your comments. We redid the article, especially the first part, because it caused most of the questions. We tried to edit the language of the article

Part 3.1 is almost completely rewritten, as it seemed not clear to both reviewers. Examples of stabilization of proteins with introduced ss-bonds are discussed in more detail.

Links and explanations on how the oxidation of ss-bonds were controlled are added (in materials and methods and in the text).

Added a link and explanations on how the ss bond affects the entropy component of the free energy of a protein.

Added links to program sites and a link to an article explaining how IsUnstruct works and some explanations in the text.

http://www.pondr.com;      

http://bioinfo.protres.ru/IsUnstruct.  

Explanations added why the experiments in Figure 8 were conducted in the presence of urea (The protein AaeL1 from extremophilic bacterium is superstable, so thermal denaturation was investigated in the presence of 4M urea)

Some additional explanations:

Referee 2

« My main concern is the confusion the Authors do between rigidity and thermodynamic stability. In other words, they observe indeed an increase of thermodynamic stability but it is quite clear that this is unrelated to the rigidity increase and it strongly depends on the introduction of a new covalent bond, … »

We tried to reformulate some phrases in the abstract and at the beginning of the article.

We hope that it has become clearer that the purpose of the work is not to find out how ss bonds affected stiffness or thermodynamic stability, but to show that experimenters can use our approach. To put it simply: Need to find a weak spot in protein? - look for a site with a high IDP! Why look for a weak spot in a protein? - it depends on the specific task. In some cases, thermal stabilization is needed, in some cases, the creation of a permutant protein.

 (2):  “In the introduction, the authors write that the “weakened” regions are stabilized, in the structured state because of the interaction with other parts of the protein. This is possible. However, one should consider the presence of crystal packing contacts.”  

- The question of why the weak parts of the protein are visible as structured in X-ray diffraction analysis is not relevant to the topic of the article.  Yes, we agree that one of the explanations is that they are “visible” due to the interaction of proteins in the crystal. On the other hand, even with the examples shown in Figure 1, it is clear that beta strands and alpha helices cannot be structured solely because of the crystal; most likely, they are similarly packaged in the native protein in solution. A striking example of GFP - it is shown that irregularities in the packaging of its beta strands lead to a loss of fluorescence.

(3):   “The strategy described in the manuscript is tested on five protein and it would be necessary to explain why these five proteins have been selected and others have not”

We rewrote this part of the article and hope that it becomes clear that we are not testing our method on five proteins. We give five proteins as an example of the fact that it is impossible to confirm our hypothesis according to published data.

(6) corresponding link is inserted

(7) the rmsd is inserted in the text.

(8) we fixed it

(9, 10)  R2: “What is new here and was not reported in reference 16 …” 

We wrote that this article brought together experimental data published by us earlier - in our opinion, this is the only way to draw generalized conclusions about the applicability of our approach.

We also fixed minor bugs and typos.

Sincerely yours,

Bogdan S. Melnik, Dr.Sc.

Laboratory of Protein Physics
Institute of Protein Research
Pushchino, Moscow Region, Russia 142290

+7 (095) 632-78-71

bmelnik@phys.protres.ru

Reviewer 2 Report

The manuscript submitted by Dr. B. S. Melnik describes a strategy to improve globular protein stability by inserting disulfide bonds in sequence moieties that have a considerable tendency to be conformationally disordered. Several experimental studies seem to support the reliability of the strategy.

My main concern is the confusion the Authors do between rigidity and thermodynamic stability. In other words, they observe indeed an increase of thermodynamic stability but it is quite clear that this is unrelated to the rigidity increase and it strongly depends on the introduction of a new covalent bond, the disulfide bond. Unless one can demonstrate that the new covalent bond does not confer stability, it is impossible to declare that the extra stability depends on the reduction of the conformational disorder.

Although potentially interesting, in my opinion, this manuscript requires considerable additional work and in its present form is NOT PUBLISHABLE.

Note also that, though I am not a native speaker, I think that some attention should be paid to improve the quality of the presentation.

Major issues

1 – Reducing the conformational disorder of some protein moieties, per se, does not guarantee an increase (in absolute value) of the folding Deta G (Gibbs free energy) since it implies an entropy reduction. An increase (in absolute value) of the folding Deta G is more likely due to the formation of an additional covalent bond (the disulfide bond). This point must be discussed.

2 – In the introduction, the authors write that the “weakened” regions are stabilized, in the structured state because of the interaction with other parts of the protein. This is possible. However, one should consider the presence of crystal packing contacts.

3 – The strategy described in the manuscript is tested on five protein and it would be necessary to explain why these five proteins have been selected and others have not.

4 – Conformational disorder predictions have been done with PONDR-FIT and IsUnstruct. There are many other predictors on the market. Why PONDR-FIT and IsUnstruct have been chosen?

5 – Section 3.1 is too superficial. If the authors cannot extend the structural bioinformatics analysis, it would be better to replace it with very few sentences. It is unclear how the data were found. Are these the only examples, in which a disulfide bond has been introduced in a globular protein? Information about secondary structure, solvent accessibility, fold type, mutations is missing.

6 – Line 204. Why these four mutant variants were selected?

7 – Superposition of AaeL1 and HmaL1: the rmsd score, the m-score, or the TM-score must be given.

8 – Protein HmaL1-E82C-D114C mod is mentioned in the caption of Figure 7 but it is not in the text. It is necessary to describe it.

9 – Section 3.4. What is new here and was not reported in reference 16?

10 – Analogously, it seems that nothing new is reported in section 3.5.

Minor issues

A – Line 34. “structural elements” à “structural domains”.

B – Last sentence of the Introduction “ … our data … were presented.” is unclear. I would say “the experimental results … are summarized.”

C – Line 254. “Spatial structure is conservative” à “The three-dimensional structure is conserved.”

D – The journal name is missing in reference 17.

Author Response

(The authors gave the same response as above.)

Round 2

Reviewer 1 Report

Manuscript is suitable for publication.

Author Response

Dear Editor and Reviewers.

Reviewer #2 states: “The revised version of the manuscript is certainly clearer. In the Abstract the sentence “This work summarizes our research on the effect of amino acid substitutions on the protein stability utilizing the outputs of the analysis of intrinsic disorder predisposition of target proteins” is sufficiently clear and per se suggests the rejection of the manuscript. A journal cannot republish what was already published by the same research group in previous articles. I suggest the Authors write a review on this topic – and this might be interesting for many readers – that cannot be confined to their studies but must cover explicitly and systematically all the (recent) publications in the field. In my opinion, this manuscript may become publishable after these modifications.”

We respectfully disagree with this statement. In fact, the reviewer's comment that in this article we are trying to publish previously published data is, to put it mildly, not true. As a matter of fact, proposed in our study methodology, where analysis of intrinsic disorder predisposition of a target protein is used for the design of its stable variants, is absolutely novel. What we are presenting here is the novel way of looking at protein to find weakest structural spots and using this information for the improvement of protein structure. My co-authors and I are convinced that the main thing in the article is an idea that needs to be confirmed or refuted by the example of different proteins and using different experimental methods. From these viewpoints, the article is completely new. It is very strange, if the reviewer believes that we can take experimental data from the articles of other authors and refer to them (the reviewer advises writing a review), but it is impossible to use our data obtained on proteins that we have been investigating for a long time.

Indeed, the proteins that are discussed in this article have been studied by us earlier, but by other methods and for completely different purposes. The article discusses the study of four proteins: GFP, GaO, L1, GroEL.

References to our previous articles are given in the manuscript, and you can see that most of our articles about GFP [18, 37-41] are devoted to the features of its intermediate states and the rates of unfolding. There is one article that actually discusses the intrinsic disorder propensity of GFP [37], but it discusses only two of the four mutant proteins described in the current manuscript.

Protein L1 and GroEL results are presented for publication for the first time. Of course, we refer to an article on GroEL published in Russian [23]. Perhaps the reviewer could not read it and accordingly did not understand that this manuscript concerns only the results of the analysis of structural properties of one of the variants of the apical domain of GroEL. This previously published paper contains neither an analysis of the intrinsic disorder propensity of this protein nor an explanation of how the design of the circular permutant was made.

Finally, although a previously published article on GaO [20] is focused on the study of the properties of this protein, the melting of its individual domains, binding to GTP, etc., it does not make an attempt to prove an idea expressed in present article (namely, using multiple proteins for validation of the proposed intrinsic disorder-based approach for designing stable protein structures).

Therefore, our article can certainly be considered a continuation of our work, but definitely not as an attempt to publish the same result in another journal.

Yours sincerely,

Bogdan Melnik   and co-authors

Reviewer 2 Report

The revised version of the manuscript is certainly clearer. In the Abstract the sentence “This work summarizes our research on the effect of amino acid substitutions on the protein stability utilizing the outputs of the analysis of intrinsic disorder predisposition of target proteins” is sufficiently clear and per se suggests the rejection of the manuscript. A journal cannot republish what was already published by the same research group in previous articles. I suggest the Authors write a review on this topic – and this might be interesting for many readers – that cannot be confined to their studies but must cover explicitly and systematically all the (recent) publications in the field. In my opinion, this manuscript may become publishable after these modifications.

Author Response

(The authors gave the same response as above.)

Round 3

Reviewer 2 Report

The pugnacious Authors of this manuscript are certainly very proud of their work, which is, in my opinion, ethically borderline, since it is absolutely unclear what is new and what is a leftover or even a replica from previous publications.

In the previous rebuttal, they write “We wrote that this article brought together experimental data published by us earlier - in our opinion, this is the only way to draw generalized conclusions about the applicability of our approach.” Now they affirm “In fact, the reviewer's comment that in this article we are trying to publish previously published data is, to put it mildly, not true.”

Some coherence would be welcome.

Note also that bullying expressions like ‘to put it mildly’ are severely inappropriate.

In this form, this manuscript cannot be published

Given that the Authors refuse the suggestion to try to write a more comprehensive review, I suggest them to try to rewrite this manuscript, by preparing a Results section, where they report only the new experimental results, and a Discussion section, where they can cite and comment previous results that are related to the new observations. In other words, by splitting the Results and Discussion section.